# Computational Modeling of Thermal Ablation Zones in the Liver: A Systematic Review

**DOI:** 10.3390/cancers15235684

**Published:** 2023-12-01

**Authors:** Gonnie C. M. van Erp, Pim Hendriks, Alexander Broersen, Coosje A. M. Verhagen, Faeze Gholamiankhah, Jouke Dijkstra, Mark C. Burgmans

**Affiliations:** 1Interventional Radiology Research (IR2) Group, Department of Radiology, Leiden University Medical Center, 2300 RC Leiden, The Netherlands; 2Division of Image Processing, Department of Radiology, Leiden University Medical Center, 2300 RC Leiden, The Netherlands

**Keywords:** thermal ablation, liver neoplasm, computational modeling, ablation zone simulation, therapy planning

## Abstract

**Simple Summary:**

Thermal ablation is an established treatment for primary and secondary liver tumors. As ablation treatment planning is a fast-emerging field, accurate and patient-specific ablation zone simulation may contribute to higher efficacy of thermal ablation. Computational modeling could facilitate these simulations. This systematic review aims to identify, evaluate, and summarize the findings of the literature on existing computational models for thermal liver ablation planning and compare their accuracy. The literature shows a wide variety of computational modeling and validation methods. Additional research, with a focus on shape-based outcome metrics, is warranted to determine which model demonstrates superior accuracy and suitability for clinical practice. More insight into parameter personalization is required to enable patient-specific ablation planning.

**Abstract:**

Purpose: This systematic review aims to identify, evaluate, and summarize the findings of the literature on existing computational models for radiofrequency and microwave thermal liver ablation planning and compare their accuracy. Methods: A systematic literature search was performed in the MEDLINE and Web of Science databases. Characteristics of the computational model and validation method of the included articles were retrieved. Results: The literature search identified 780 articles, of which 35 were included. A total of 19 articles focused on simulating radiofrequency ablation (RFA) zones, and 16 focused on microwave ablation (MWA) zones. Out of the 16 articles simulating MWA, only 2 used in vivo experiments to validate their simulations. Out of the 19 articles simulating RFA, 10 articles used in vivo validation. Dice similarity coefficients describing the overlap between in vivo experiments and simulated RFA zones varied between 0.418 and 0.728, with mean surface deviations varying between 1.1 mm and 8.67 mm. Conclusion: Computational models to simulate ablation zones of MWA and RFA show considerable heterogeneity in model type and validation methods. It is currently unknown which model is most accurate and best suitable for use in clinical practice.

## 1. Introduction

Percutaneous thermal ablation is an established minimally invasive treatment for primary and secondary liver tumors [1,2]. Radiofrequency ablation (RFA) and microwave ablation (MWA) are currently the most widely applied thermal ablation techniques to treat liver malignancies. Both techniques aim to induce tissue heating of at least 55–60 °C to necrotize the tumor along with an ablative margin of normal liver parenchyma around the tumor of at least 5 mm [3,4]. RFA applies a rapidly alternating current that excites the ions in the liver tissue, causing frictional heating. In MWA, electromagnetic waves cause polar molecules, predominantly water, to realign with the oscillating field, which generates heat through kinetic energy [5].

The obtained ablative margin in thermal ablation is correlated with local recurrence rates. In a study by Laimer et al. conducted on patients with hepatocellular carcinoma, each millimeter increase in the minimal ablative margin resulted in a 30% risk reduction for local recurrence [4]. No recurrences occurred when an ablative margin of >5 mm was obtained, but this was only achieved in 37.5% of the ablations. These results are in accordance with several other studies investigating the correlation between ablative margin and local recurrences in primary and secondary liver tumors [3,4,6,7,8,9]. In these studies, the percentage of the intended ablative margin of >5 mm varied between 2.7% and 51.4%. These low rates indicate a discrepancy between the predicted and created ablation zone, either due to shape and size deviation of the created ablation zone or inaccurate needle positioning.

Currently, the ablation zone size is predicted according to the manufacturer’s specifications. With each system, a chart is provided containing 2D ellipse predictions for several settings (such as ablation time and power). In recent years, treatment planning tools have become available that use these predictions to help the operator choose the probe trajectory and ablation setting [10]. Yet, these predictions are mostly based on preclinical animal experiments [11]. Various factors, such as vascular proximity, tissue perfusion, tumor location, and underlying fibrosis or cirrhosis, can lead to differences in the shape and volume of the ablation zone compared to predictions. Computational modeling and ex vivo experiments have demonstrated that the aforementioned tumor and liver characteristics affect the heat conductivity and thus the dimensions of the actual ablation zone [12,13,14,15]. To increase the rate of adequate ablations with sufficient margins and thereby reduce the risk of local recurrence, patient-specific therapy planning is essential.

The computation of ablation necrosis volumes generally consists of two steps. First, increase in temperature during thermal ablation can be simulated by computational models based on electromagnetic and bioheat equations [16,17]. Cell-death models then convert these temperatures to a volume of necrosis. A wide variety of complexity is observed in these models [16,17,18]. Whereas some computational models rely only on a bioheat equation with fixed parameters, other models incorporate more complexities, such as patient-specific anatomy and temperature-dependent parameters.

Computational models are currently mainly used for in silico testing and optimization of devices as well as to study the effect of different parameters and techniques on ablation treatment [19,20,21]. As treatment planning is a fast-emerging field, accurate and patient-specific ablation zone simulation may contribute to higher efficacy of thermal ablation [10]. Computational modeling potentially opens up an opportunity for patient-specific ablation zone simulation. This systematic review aims to identify, evaluate, and summarize the findings of the literature on existing computational models for radiofrequency and microwave thermal liver ablation planning and compare their accuracy.

## 2. Materials and Methods

### 2.1. Search Strategy

Studies were identified by searching the electronic databases MEDLINE and Web of Science on 17 April 2023. The search queries were based on synonyms of the keywords “Thermal ablation”, “Liver neoplasm”, and “Computational modeling”. The complete search strategies used can be found in Appendix A. The study has not been registered in PROSPERO.

### 2.2. Study Selection

After duplicate removal, abstracts were screened, followed by full-text assessment. Articles were found eligible if (i) ablation zone simulation was performed for either (ii) percutaneous RFA or MWA in (iii) liver tissue and if (iv) the model was quantitatively validated using ex vivo or in vivo experiments with (v) either ablation zone dimensions, ablation zone volumes, or volume-based metrics reported as outcome measure(s). Furthermore, articles should focus on simulating ablation zones for treatment planning and not for device or protocol optimization. Reviews, systematic reviews, letters to the editor, and articles written in other languages than English were excluded. Two authors (G.C.M.v.E. and P.H.) independently assessed the articles according to these criteria. In case of disagreement, consensus was reached by discussion.

### 2.3. Data Extraction

Included articles were categorized based on the thermal ablation technique used, either RFA or MWA. For each included computational model, the applied biological heat transfer model and the cell death model were extracted. Furthermore, it was noted whether perfusion, blood vessels, water vaporization, temperature-dependent thermal parameters, and/or an image-based anatomical model were incorporated. Data extraction for the validation of computational models included study type (in vivo, ex vivo, or clinical studies), number of ablations, ablation settings, ground truth comparison, outcome metrics, and validation results. The relative volume deviation (RVD) or relative diameter deviation (RDD) between the simulated and experimentally obtained ablation zones was calculated based on reported ablation zone volumes or diameters, if this outcome was not reported on already, to ensure homogenous size-based outcome metrics.

## 3. Results

### 3.1. Study Selection

The search strategy identified 849 articles after removal of duplicates. A total of 673 articles were excluded after abstract screening. A total of 176 articles were full-text assessed, resulting in the exclusion of 141 articles. A total of 35 articles met the inclusion criteria and were included in this systematic review [22,23,24,25,26,27,28,29,30,31,32,33,34,35,36,37,38,39,40,41,42,43,44,45,46,47,48,49,50,51,52,53,54,55,56]. Figure 1 shows a flow diagram of the study-selection process.

Out of the 35 articles included, 16 focused on simulating MWA and 19 on RFA. Table 1 and Table 2 contain details about the computational model used in these articles. Figure 2 gives a schematic overview of a general model structure. Frequently used equations in the models can be found in Appendix B and frequently used outcome metrics in Appendix C.

### 3.2. MWA Data Analysis

Sixteen articles presented computational models for MWA. Thirteen articles used Pennes’ bioheat equation as bioheat model [26,29,30,35,36,37,38,44,46,52,53,55,56], while one article used the transient heat transfer equation [33], one the local thermal non-equilibrium equation [48], and one used a proprietary heat transfer model [31]. As a cell death model, the Arrhenius thermal damage model was used in four articles, the three-state cell death equation was used in two articles, and an isothermal contour (54 °C or 60 °C) was used in eight articles. Two articles used both the Arrhenius thermal damage model and an isothermal contour of 52 °C and 54 °C, respectively [30,53].

Seven articles included perfusion in their model, two blood vessels, thirteen water vaporization, and thirteen temperature-dependent tissue parameters. Two models used CT-based anatomy models. Gao et al. used CT data to extract tumor geometry to model tumor coverage, while Zhai et al. created a complete CT-based 3D model for simulating the ablation [36,56].

#### 3.2.1. MWA Ex Vivo Validation

Table 3 gives an overview of the ex vivo validation of the included computational models for MWA simulation. Fourteen articles used ex vivo experiments in animals or phantoms [26,29,30,31,33,35,36,37,38,44,46,52,53,55]. Two articles used the Dice similarity coefficient (DSC) to express their results and found similar scores between 0.74 and 0.82 [31,33]. One article used the Jaccard similarity index and found results of 0.866 and 0.934. However, these results might be biased since the electrical and thermal conductivities were reconstructed after the experiments to best fit the model [29]. Sing et al. used the experiments of Wu et al. to validate their simulated ablation zone [44,55]. The main differences between the two models were the use of the three-state cell death model and the incorporation of tissue shrinkage within the model of Sing et al. The latter could explain why simulations by Sing et al. resulted in a smaller longitudinal diameter compared to Wu et al.: 26.24 mm (RDD: −13.4%) and 29.7 mm (RDD = −2.0%), respectively. However, the experiments of Sing et al. resulted in a greater overestimation of the transverse diameter (RDD: 5.2% versus 4.7%). Figure 3 visually gives an overview of the models validated with the longitudinal and transverse RDD.

#### 3.2.2. MWA In Vivo Validation

Table 4 presents an overview of the two articles that used in vivo validation in patients [48,56]. Tucci et al. modeled four different blood vessels and compared them to the in vivo experiments of Amabile et al. [48,58]. They concluded that their model, including terminal arteries, showed a good agreement with the ablation zones achieved in the clinical study. Zhai et al. performed a study on nine patients [56]. Ablation simulation had an RVD of ±7.0% compared to clinically obtained ablation volumes. However, the study has a limited sample size without uniform image analysis.

### 3.3. RFA Data Analysis

Nineteen articles presented computational models for RFA. Thirteen articles used Pennes’ bioheat equation as a bioheat model [25,27,28,32,34,39,40,41,42,45,49,50,51], while one article used the heat transfer equation [54], one the split volume bioheat equation [43], and one article compared three different bioheat-models, i.e., Pennes’ bioheat equation, the local thermal equilibrium equation, and the local thermal non-equilibrium equation [47]. Three articles by Audigier et al. used a combination of Pennes’ bioheat equation with the Wulff–Klinger model [22,23,24]. As a cell death model, the Arrhenius thermal damage model was used in eight articles, the three-state cell death equation in seven, and an isothermal contour in three articles. Subramanian et al. used their own thermal damage formula [45]. Thirteen articles included perfusion in their model, twelve articles included blood vessels, six included water vaporization, and eight included temperature-dependent tissue parameters. Nine articles created a CT-based anatomical model for their simulation. Eight of them segmented the liver, tumor, and blood vessels, while Ooi et al. only derived a liver contour from the CT scan [42]. Next to an anatomical model, Moche et al. used dynamic CT measurements to derive perfusion values [41].

#### 3.3.1. RFA Ex Vivo Validation

Table 5 contains the ex vivo validation of the RFA ablation zone models [27,28,32,34,42,45,49,51,54]. All experiments were performed on animal livers, either bovine or porcine, or phantoms. The experimental ablation zones were measured after tissue sample sections along the probe axis. The ablation settings differed in all experiments. A visual overview of the longitudinal and transverse RDD is given in Figure 4. Figure 5 contains a combined overview of the MWA and RFA ex vivo experiments.

#### 3.3.2. RFA In Vivo Validation

An overview of the ten articles on RFA ablation zone simulation using in vivo validation is given in Table 6. These consist of four in vivo animal experiments [24,39,43,47], five retrospective clinical studies [22,23,25,40,50], and one prospective clinical study [41]. The prospective study of Moche et al. found a DSC of 0.62 ± 0.14 with a surface deviation of 3.4 ± 1.7 mm [41]. They concluded that the real-time simulation of RFA-induced tissue necrosis in the liver was fast (3.5 ± 1.9 min) and accurate enough for therapy planning. Mariappan et al. used the same computational model in their retrospective study and found similar results in 23 ablations, with a slightly lower surface deviation (2.50 mm versus 3.4 mm) [40]. The simulation accuracy increased by using patient-specific CT-based perfusion values. The results obtained by Audigier et al. indicated lower accuracy of their simulation model compared to the previously mentioned studies [22,23,24]. They found a DSC of 0.44 and a surface deviation of 5.3 ± 3.6 mm [24]. This difference in results might be explained by the reconstruction of the ablation probe location. Audigier et al. did not reconstruct the clinically used probe location but assumed the center of the tumor as the probe location in their simulation, which introduces an inaccuracy in the measurements. On the other hand, Moche et al. and Mariappan et al. reconstructed the probe location as used in clinical practice by using image registration [40,41].

Two articles compared their computational model to more simplistic models [25,39]. Hoffer et al. demonstrated an improved accuracy for computational models compared to the manufacturers’ chart, with mean surface deviations of 1.1 mm and 2.5 mm, respectively [39]. Audigier et al. also found that patient-specific models resulted in higher DSC and surface deviations compared to simplistic ellipse models [25].

## 4. Discussion

A wide variety of computational modeling and validation methods was found in this systematic review, making a full-fledged comparison between different models for ablation simulation complex. The studies included in this review used different bioheat equations, parameters, validation methods, and outcome measures or metrics, making it hard to draw conclusions on which computational model performs best and could possibly be implemented in clinical practice.

Considerable differences were identified in the RFA and MWA models, which can be explained by the different heating mechanisms of the two techniques. Thirteen of the sixteen (81%) MWA models included the effect of water vaporization and temperature-dependent parameters, compared to six of the nineteen RFA models (32%). The temperature-dependency of dielectric parameters (electrical conductivity and permittivity) and thermal conductivity is two-folded. First, tissue parameters change as a result of protein denaturation at temperatures above 60 °C [60,61,62]. In addition, water vaporization has an isolating effect. This leads to a decrease in conductivity and permittivity above 100 °C in RFA as well as in MWA [60,61,62]. However, RFA power is usually temperature- or impedance-controlled to avoid vaporization and gas formation, whereas tissue temperatures in MWA frequently exceed 100 °C. Therefore, temperature-dependent parameters, as well as water vaporization, have a greater potential influence on MWA. This could explain the discrepancy in incorporating these parameters in MWA models and RFA models. Moreover, blood vessels are included more frequently in RFA models compared to MWA models (12 out of 19 (63%) and 2 out of 16 (12.5%), respectively). Since the “heat-sink effect” caused by blood vessels near the ablation region is considered to have a greater impact on RFA, this finding seems logical [63,64]. When designing an ablation simulation model, these differences in the heating mechanism should be taken into consideration.

The various models applied different bioheat equations, which is the basis for the simulation and therefore affects the simulation accuracy. The majority of the models are purely based on Pennes’ bioheat equation due to its simplicity and feasibility. Nevertheless, this equation has an important limitation. The equation only considers microvascular perfusion, assuming a constant blood temperature of 37 °C without flow directionality. However, the blood temperature of vessels within and surrounding the ablation zone will increase during ablation [42]. Other bioheat equations that incorporate changes in liver perfusion overcome this limitation while maintaining or even increasing model performances [47,48]. In the liver, which is a well-perfused organ with variations in perfusion due to underlying liver diseases, these more advanced bioheat equations are likely to be more accurate.

In addition to the bioheat equation, there are numerous other model design choices affecting the simulation accuracy. This systematic review does not present all possible characteristics of the models. For example, it does not incorporate tissue contraction or two-compartment models. The latter models the tumor and liver as separate compartments, each with its own tissue-specific parameters. This limitation emphasizes the complexity of ablation zone prediction and the number of parameters potentially affecting the ablation zone. Theoretically, the highest accuracy could be achieved by the inclusion of all model characteristics and parameters, but there is a trade-off between accuracy on one hand and complexity and required computational resources on the other. With the recent advancement in artificial intelligence and machine learning, the development of more complex and accurate models becomes feasible.

The chosen input parameter values required for the computational model also contribute to the accuracy achieved. Some input parameters, e.g., thermal conductivity, electrical conductivity, and density, are dependent on tissue composition. They differ between tumor and liver parenchyma as well as per patient as a result of underlying liver disease, i.e., cirrhosis, hepatic steatosis, and previous treatments like systemic treatment or transarterial radioembolization or chemoembolization.

Several studies concluded that thermal conductivity significantly decreases with an increasing fat content [29,65,66,67]. According to the computational models of Deshazer et al. and Servin et al., a greater fat content in the liver leads to larger ablation zone volumes in MWA [66,67]. Due to the lower conductivity of surrounding liver parenchyma, heat loss to peripheral parenchyma is limited since low thermally conducting tissue retains high temperatures. This leads to larger ablation zones in MWA. RFA relies more on indirect heating, i.e., heat conductivity, compared to direct heating in MWA. Therefore, smaller ablation zones are observed in livers with increasing fat content. Diminished conductivity of the parenchyma results in higher temperatures at the site of the tumor edge, while the liver parenchyma surrounding the tumor has a lower temperate increase [65]. Hypothetically, this “oven-effect” would lead to higher ablation temperatures within the tumor while increasing the risk of narrow ablative margins in RFA. In vivo experiments are needed to test these hypotheses.

In contrast to the widely studied effect of hepatic steatosis in thermal liver ablation, limited research has been conducted on the relation between ablation volumes and cirrhosis or fibrosis. This might be related to the complexity of parenchymal changes due to cirrhosis and fibrosis. Deshazer et al. simulated the difference in perfusion between normal liver tissue and cirrhotic liver tissue and found larger ablation zones in cirrhotic livers compared to normal liver tissue [67]. Further studies are needed to investigate heat propagation and conductivity in cirrhotic and fibrotic liver tissue to be able to include these as patient-specific input parameters in liver ablation simulation.

The simulation accuracy is determined by model validation. The included studies varied in validation methods using in or ex vivo experiments with different outcome measures or metrics. Although the bottleneck in current practice is that the manufacturer’s prediction is mostly based on ex vivo experiments, 23 out of 38 included models are still validated with ex vivo experiments [11]. Moreover, in most studies, the axis length and total volume of the simulated and actual ablation are compared without taking into account the shape and relative position. These variations and limitations in the model validation make it difficult to compare the different simulation models and to identify model parameters that have the largest impact on model accuracy.

To decide on the optimal balance between complexity, accuracy, and computational time, a comparative clinical study should be conducted. This could be a retrospective or prospective study in which the clinically obtained ablation zone is compared to different simulations. This validation necessitates a probe position scan to simulate the ablation at the corresponding probe position, as well as a post-ablation CT scan to segment the clinically obtained ablation zone. To compare the shape and position of the simulated and clinically obtained ablation zone, image registration of the post-ablation and probe position CT scan is required. Variations in liver shape and position should be minimized by identical patient positioning and breathing phase in both scans, which can be controlled with techniques such as high-frequency jet ventilation [68]. Nevertheless, image registration remains a challenge due to ablation-induced deformations of the liver. Therefore, registration might introduce inaccuracies to the measurements. After image registration, volumetric outcome metrics, such as the DSC and surface deviations, could be used to compare the overlap of the simulated and clinically obtained ablation zones. These outcome metrics are more expressive in clinical practice compared to size-based metrics since the ablation zone should cover the tumor, including an ablative margin of at least 5 mm. Therefore, the simulation should not only be an accurate prediction of size and volume but also of shape and position relative to the ablation probe. In addition, the maximum surface deviation should be computed since it is indicative of the boundary discrepancies as well as the negative maximum surface deviation and the false positive value (Appendix C, Figure A2c,g), which might be a good measure for the technical success of the ablation. The different models could be compared to determine which combination of included parameters and models results in the most accurate prediction.

The clinical application of ablation zone simulation requires reliable models. Three of the included articles originate from the ClinicIMPPACT project, which aims to bring a planning and simulation tool for RFA into the clinical practice [40,41,50,69]. In a clinical study, Moche et al. evaluated this application prospectively and concluded that the model performs sufficiently for clinical implementation with a mean surface deviation of 3.4 mm [41]. Nevertheless, it is generally recommended to aim for an ablative margin of at least 5 mm, but the exact relationship between ablative margins and local recurrence is still unknown [70]. Therefore, it is questionable if a mean surface deviation of 3.4 mm provides sufficient accuracy. Furthermore, to conclude on the accuracy needed for clinical implementation of computational models, the reliability of the current clinical practice, i.e., the manufacturer’s prediction, should be known. Hence, the match between the manufacturer’s prediction and the clinically obtained ablation zone should be quantified. Of interest is a study by Hoffer et al. They compared their computational model with the manufacturer’s chart and performed in vivo ablations in six swine and found their computational model to have higher accuracy [39]. However, clinical studies with a large patient population are still lacking.

Percutaneous thermal ablation techniques have evolved over the most recent decades with new innovations. One advancement is the use of treatment planning software combined with probe navigation using either stereotactic thermal ablation or robotic probe positioning to increase thermal ablation efficacy. However, this planning software still relies on the manufacturer’s predicted ablation zones, lacking patient-specific parameters. The full potential of this innovation can only be exploited when reliable patient-specific ablation zone predictions are available [10]. The use of image-based two-compartment models may be an important step towards a more accurate, patient-specific model. The comparative study of Audigier et al. has indicated that patient-specific modeling results in more accurate predictions compared to a spherical model [25]. Moreover, Mariappan et al. concluded that the simulation yields better accuracy when personalized perfusion values are given as input in the simulation model [40]. These values can be extracted from CT images, which also enables two-compartment modeling that incorporates the liver vasculature. These results emphasize the potential of computational modeling for patient-specific ablation planning. However, more insight into parameter personalization is required to enable patient-specific ablation planning and implement this into clinical practice.

## 5. Conclusions

Computational models simulating ablation zones in MWA and RFA show considerable heterogeneity in model type and validation methods. It is currently unknown which model is most accurate and best suitable for use in clinical practice. However, several studies have demonstrated a good correlation between simulated ablation zones and in vivo ablations. More research on patient-specific parameters is needed to develop more accurate models that can be used for individualized treatment planning.

## Figures and Tables

**Figure 1 cancers-15-05684-f001:**
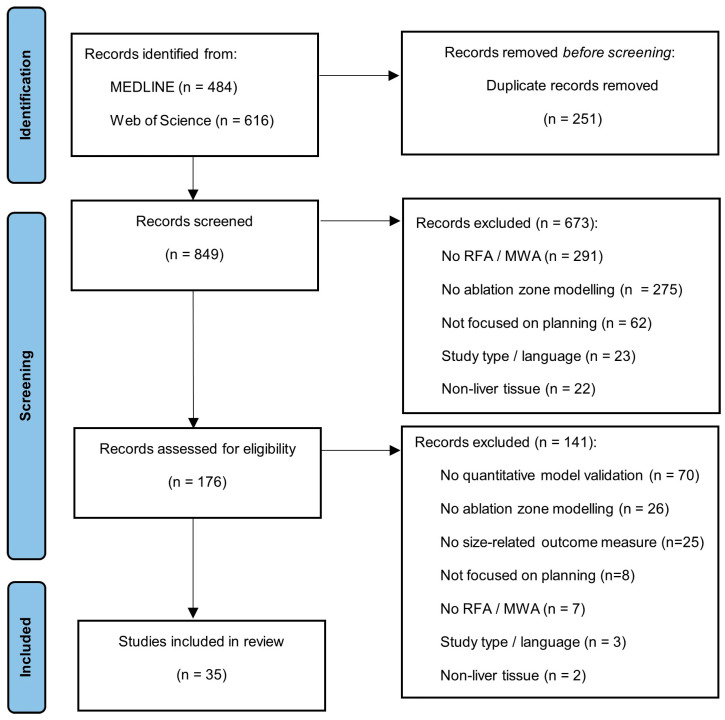
Preferred Reporting Items for Systematic Reviews and Meta-Analysis (PRISMA) flow diagram describing the study-selection process.

**Figure 2 cancers-15-05684-f002:**
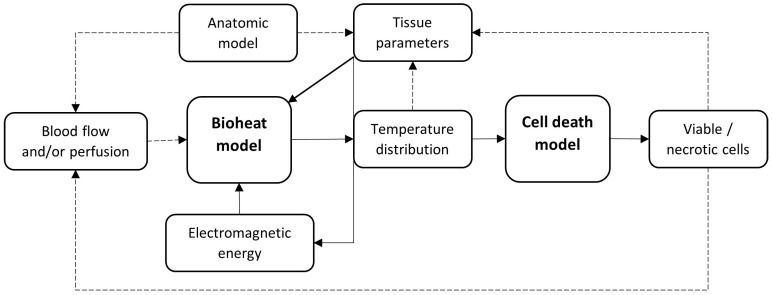
Schematic overview of the basic structure of a computational model for simulating thermal liver ablation. The optional dependencies are shown by the dotted lines.

**Figure 3 cancers-15-05684-f003:**
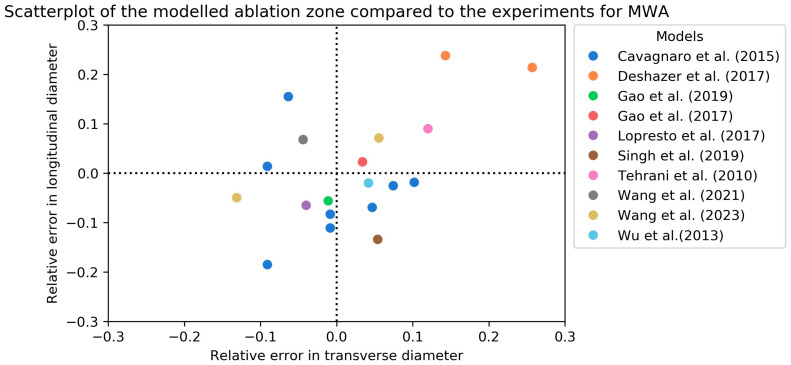
Scatterplot of the relative error in longitudinal and transverse diameters of the modeled MWA zones compared to ex vivo validation [26,30,35,37,38,44,46,52,53,55]. In case of an experimental diameter of 30 mm, a relative error of 0.1 means the simulated diameter was 33 mm.

**Figure 4 cancers-15-05684-f004:**
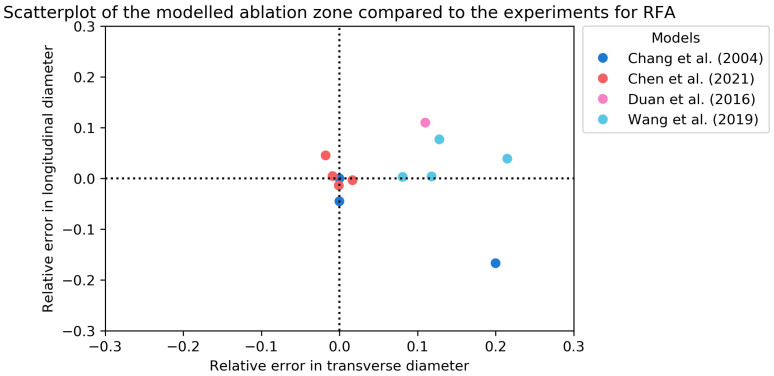
Scatterplot of the relative error in longitudinal and transverse diameters of the modeled RFA zones compared to ex vivo validation [27,28,32,51]. In case of an experimental diameter of 30 mm, a relative error of 0.1 means the simulated diameter was 33 mm.

**Figure 5 cancers-15-05684-f005:**
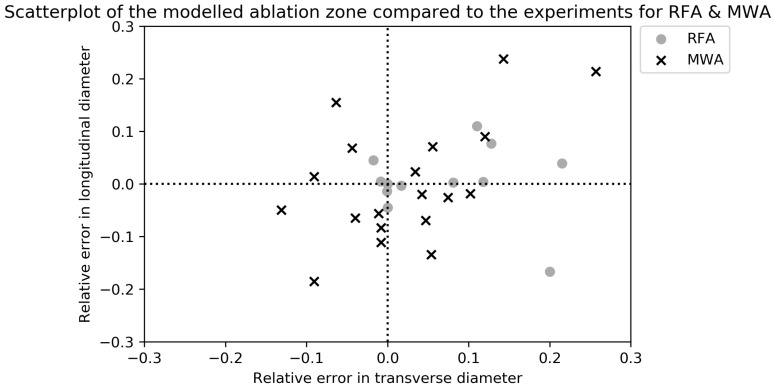
Scatterplot of the relative error in longitudinal and transverse diameters of the modeled MWA and RFA zones compared to ex vivo validation. In case of an experimental diameter of 30 mm, a relative error of 0.1 means the simulated diameter was 33 mm.

**Table 1 cancers-15-05684-t001:** Characteristics of the used computational model for microwave ablation zone modeling from the included articles (x = incorporated in the model).

Author (Year)	Bioheat Model	Cell Death Model	Numerical Method *	Perfusion	Blood Vessels	Water Vaporization	Temperature-Dependent Tissue Parameters	CT-Based Anatomic Model	Model Remarks
Cavagnaro et al. [26] (2015)	Pennes’ BHE	60 °C isothermal contour	FDTD						BHE-S: Standard BHE
	Pennes’ BHE	60 °C isothermal contour				x			BHE-V: Standard BHE including water vaporization
	Pennes’ BHE	60 °C isothermal contour					x		BHE-ST_B and BHE-ST (two different equations for temperature-dependent parameters)
	Pennes’ BHE	60 °C isothermal contour				x	x		BHE-V-ST_B and BHE-V-ST. (two different equations for temperature-dependent parameters), only conductivity is temperature-dependent
	Pennes’ BHE	60 °C isothermal contour				x	x		SAR-T-1min_B and SAR-T-1min (two different equations for temperature-dependent parameters). Temperature-dependency of conductivity as well as dielectric parameters
Collins et al. [29] (2020)	Pennes’ BHE	Arrhenius thermal damage model	FEM						Determine dielectric properties based on MRI fat quantification with inverse-modeling strategy
Deshazer et al. [30] (2017)	Pennes’ BHE	Arrhenius thermal damage model (isocontour 63%) and 52 °C isothermal contour	FEM	x, but not in experiments		x	x		Damage-dependent blood perfusion rate. Two different models tested (A and B); they only differ in dielectric parameter dependency of temperature
Deshazer et al. [31] (2017)	Own heat-transfer model	60 °C isothermal contour	FEM			x	x		Investigated the option of intra-procedural SAR measurement to model ablation zone
Faridi et al. [33] (2020)	Transient heat-transfer equation	Arrhenius thermal damage model (isocontour 63%)	FEM			x	x		Added the Morris method to determine the sensitivity of the ablation zones to uncertainty in tissue physical properties
Gao et al. [35] (2017)	Pennes’ BHE	54 °C isothermal contour	FEM						Used experiments to determine phantom parameters and SAR distribution, which is the basis of the FEM model
Gao et al. [36] (2019)	Pennes’ BHE	54 °C isothermal contour	FEM			x	x	x	Tried to model coagulation zone over time and incorporate tumor geometry to assess tumor coverage
Gao et al. [37] (2019)	Pennes’ BHE	54 °C isothermal contour	FEM			x	x		Used parameter sensitivity analysis to optimize the temperature-based parameters
Lopresto et al. [38] (2017)	Pennes’ BHE	60 °C isothermal contour	FDTD			x	x		Evaluate the effect of ±25% variations in dielectric and thermal parameters using the combined expanded uncertainty
Singh et al. [44] (2019)	Pennes’ BHE with Dual phase lag model	Three-state cell death model	FEM	x, but not in experiments		x	x		Incorporates lot of complexities: damage-dependent blood perfusion rate, mechanical deformation (shrinkage) and heat-flux model. Modeled RFA as well as MWA. However, only validated MWA with experiments
Tehrani et al. [46] (2010)	Pennes’ BHE	Three-state cell death model	FEM	x		x			Used a multicompartment model including tissue, tumor and blood. Added a model for tumor shrinkage
Tucci et al. [48] (2022)	Local thermal non-equilibrium equation	Arrhenius thermal damage model (isocontour 99%)	FEM	x	x, 4 different diameters	x	x		Damage-dependent blood perfusion rate. Two compartment model with difference in porosity (and other factors) in tumor and surrounding liver tissue. Also, within the tumor, the difference in porosity in the tumor core toward the tumor rim (increasing porosity) is modeled
Wang et al. [52] (2021)	Pennes’ BHE	54 °C isothermal contour	FEM	x	x	x	x		Incorporated convection heat-transfer condition and Newton formula for heat transfer between blood vessel and tissue
Wang et al. [53] (2023)	Pennes’ BHE	54 °C isothermal contour and Arrhenius thermal damage model (isocontour 63%)	FEM	x, but not in experiments		x	x		Modeled dual-antenna MWA, different distances between antennas
Wu et al. [55] (2013)	Pennes’ BHE	55 °C isothermal contour	FDTD			x	x		Used GPUs to simulate in 3D. Did not quantify the electrical field, but determined its contribution based on experiments.
Zhai et al. [56] (2008)	Pennes’ BHE	Arrhenius thermal damage model (isocontour 63%)	FEM	x			x	x	GPU-accelerated model for preoperative 3D simulation of necrotic zone in clinical setting. Incorporated effect of necrosis on blood perfusion

* FDTD = Finite Difference Time Domain, FEM = Finite Element Method, FVM = Finite Volume Method. BHE = bioheat equation, CT = Computed Tomography, GPU = graphics processing unit, MWA = microwave ablation, SAR = specific absorption rate.

**Table 2 cancers-15-05684-t002:** Characteristics of the used computational model for radio-frequency ablation zone modeling from the included articles (x = incorporated in the model).

Author (Year)	Bioheat Model	Cell Death Model	Numerical Method *	Perfusion	Blood Vessels	Water Vaporization	Temperature-Dependent Tissue Parameters	CT-Based Anatomic Model	Model Remarks
Audigier et al. [22] (2013)	Combination of Pennes’ BHE and Wulff–Klinger model	Three-state cell death model	Lattice Boltzmann solver	x	x			x	Computational fluid dynamics and Darcy’s equation are coupled to the bioheat equation to model blood circulation and blood flow
Audigier et al. [23] (2015)	Combination of Pennes’ BHE and Wulff–Klinger model	Three-state cell death model	Lattice Boltzmann solver	x	x			x	Computational fluid dynamics and Darcy’s equation are coupled to the bioheat equation to model blood circulation and blood flow, two-compartment model (blood vessels and liver tissue)
Audigier et al. [24] (2017)	Combination of Pennes’ BHE and Wulff–Klinger model	Three-state cell death model	Lattice Boltzmann solver	x	x			x	Navier-stokes equation and computational fluid dynamics solver used to model blood flow. Blood flow is determined using preoperative MRI, blood pressures are measured invasively, and porosity map created on CT image. Used intra-operative measurements to validate parameter values used. Used lower conductivity for cirrhotic livers
Audigier et al. [25] (2022)	Pennes’ BHE	50 °C isothermal contour	Lattice Boltzmann solver		x			x	Also used a spherical model and Eikonal model for comparison. Used a GPU for acceleration, multi-probe modeling
Chang et al. [27] (2004)	Pennes’ BHE	Arrhenius thermal damage model (isocontour 63%)	FEM	x, but not in experiments			x		Damage-dependent blood perfusion ratel
Chen et al. [28] (2021)	Simplified Pennes’ BHE	55 °C isothermal contour	Simplified toward analytical solution						Ignored the heat source of the electrical current flow in the model
Duan et al. [32] (2016)	Pennes’ BHE	Arrhenius thermal damage model (isocontour 63%)	FEM	x, but not in experiments			x		Using a pre-procedural determined probe position; the probability of several ablation zones is displayed by the model. Damage-dependent blood perfusion ratel
Fang et al. [34] (2022)	Pennes’ BHE	Arrhenius thermal damage model (isocontour 99%)	FEM	x, but not in experiments	x	x	x		Used the Navier–Stokes equation for blood flow modeling
Hoffer et al. [39] (2022)	Pennes’ BHE	Arrhenius thermal damage model (isocontour < 63%)	FEM and FDM	x	x	x			Used a GPU to accelerate FEM, able to model single and multi-probe ablations, focused on clinical application
Mariappan et al. [40] (2017)	Pennes’ BHE	Three-state cell death model	FEM	x	x			x	Used a GPU to accelerate FEM, focused on clinical application
Moche et al. [41] (2020)	Pennes’ BHE	Three-state cell death model	FEM	x	x			x	Used a GPU, more focused on clinical application. Simulation parameters involved a proportional integral derivative
Ooi et al. [42] (2019)	Pennes’ BHE	Arrhenius thermal damage model (isocontour 99%)	FEM	x, but not in experiments		x	x	x	Modeled different boundary conditions
Payne et al. [43] (2011)	Split-volume bioheat equation (own model)	Three-state cell death model	FEM	x	x			x	Incorporated Newton’s cooling law to model heat transfer between vessels and tissue and Darcy’s law for blood velocity
Subramanian et al. [45] (2015)	Pennes’ BHE	Own thermal damage formula	FEM				x		Experimental-based values of the specific heat, thermal conductivity, and electrical conductivity
Tucci et al. [47] (2021)	Pennes’ BHE	Arrhenius thermal damage model (isocontour 99%)	FEM	x		x			Damage-dependent blood perfusion rate
	Local thermal equilibrium equation	60 °C isothermal contour		x		x			Porous media-based model, damage-dependent blood perfusion rate; assumes equilibrium in temperature between blood and tissue
	Local thermal non-equilibrium equation	60 °C isothermal contour		x		x	x		Porous media-based model, damage-dependent blood perfusion rate; separates vaporization phase for water, tissue, and blood
Vaidya et al. [49] (2021)	Pennes’ BHE	Arrhenius thermal damage model	FVM		x	x	x		Multicompartment model incorporating tissue, tumor, blood, and probe. Damage-dependent blood perfusion rate
Voglreiter et al. [50] (2018)	Pennes’ BHE	Three-state cell death model	FEM	x	x			x	Used a GPU to accelerate FEM; focused on clinical application
Wang et al. [51] (2019)	Pennes’ BHE	54 °C isothermal contour	FEM				x		
Welp et al. [54] (2006)	Heat transfer equation	Arrhenius thermal damage model (isocontour 99%)	FEM		x	x	x		Incorporated the heat transfer between blood and tissue

* FDTD = Finite Difference Time Domain, FEM = Finite Element Method, FVM = Finite Volume Method, FDM = Finite Difference Method. BHE = bioheat equation, CT = Computed Tomography, GPU = graphics processing unit, RFA = radio-frequency ablation.

**Table 3 cancers-15-05684-t003:** Ex vivo validation of computational models modeling microwave ablation.

Author (Year)	Model	In Vivo or Ex Vivo Validation	Number of Experiments	Ground Truth	Ablation Settings (Time of Ablation and Power)	Outcome Measure/Metric	Performance	Validation Remarks
Cavagnaro et al. [26] (2015)	BHE-S	Ex vivo, bovine livers	6	Sectioning sample and measure ablation zone	10 min, 40 W, 2450 MHz	Longitudinal (L) and transverse (T) RDD *	L: −8.31% T: −0.83%	
	BHE-V						L: −18.5% T: −9.09%	
	BHE-ST_B						L: −1.85% T: 10.2%	
	BHE-ST						L: −2.54%, T: 7.44%	
	BHE-V-ST_B						L: −6.93%, T: 4.68%	
	BHE-V-ST						L: −11.1%, T: −0.83%	
	SAR-T-1min_B						L: 15.5%, T: −6.34%	
	SAR-T-1min						L: 1.39%, T: −9.09%	
Collins et al. [29] (2020)	Fat phantoms	Ex vivo, phantom	15	Sectioning sample, photographed and 2D segmentation of ablation zone	15 min, 60 W, 915 MHz	Jaccard similarity index	0.866 ± 0.053	For each phantom, the electrical and thermal conductivity were reconstructed to best fit the model
	Non-fat phantom		6			Jaccard similarity index	0.934 ± 0.022	
Deshazer et al. [30] (2017)	Model A	Ex vivo, bovine livers	4	Sectioning sample and measure ablation zone	10 min, 30 W, 915 MHz	Longitudinal (L) and transverse (T) RDD *	L: 2.9%, T: 24.0%	A: linear temperature dependency of dielectric properties, B: similar to model A but added linear decrease in electrical conductivity above 95 °C
	Model B		4				L: 5.7%, T: 12.0%	
	Model A		8		15 min, 60 W, 915 MHz		L: 21.4%, T: 25.7%	
	Model B		8				L: 23.8%, T: 14.3%	
Deshazer et al. [31] (2017)	Short-tip, 1000 W/kg iso-SAR	Ex vivo, porcine livers	3	Segmentation on infrared camera temperature measurements	6 min, 15 W, 915 MHz	DSC	0.74 ± 0.01	
	Short-tip, 500 W/kg iso-SAR						0.82 ± 0.04	
	Long-tip, 1000 W/kg iso-SAR		3				0.77 ± 0.03	
	Long-tip, 500 W/kilo iso-SAR						0.76 ± 0.01	
Faridi et al. [33] (2020)		Ex vivo, bovine livers	4	Segmentation on MRT-derived Arrhenius thermal damage 3D maps	10 min, 30 W, 2450 MHz	DSC	0.8 ± 0.0	
			8		5 min, 30 W, 2450 MHz		0.8 ± 0.08	
			3		5 min, 50 W, 2450 MHz		0.75 ± 0.06	
Gao et al. [35] (2017)		Ex vivo, phantom		Sectioning sample and measure ablation zone	10 min, 60 W, 2450 MHz	Longitudinal (L) and transverse (T) RDD *	L: −5.6%, T: −1.1%	
						Advancement	0.341 vs. 0.3 ± 0.05 cm	
Gao et al. [36] (2019)		Ex vivo, porcine livers	20	Sectioning sample and measure ablation zone	40, 45, 50, 55 and 60 W, 2450 MHz	Error of transverse radius, advancement and backward longitudinal length	±5%	
Gao et al. [37] (2019)		Ex vivo, porcine livers	20	Sectioning sample and measure ablation zone	6 min, 60 W, 2450 MHz	Longitudinal (L) and transverse (T) RDD *	L: 2.3%, T 3.4%	Optimized thermo-dependent parameters based on experiments
Lopresto et al. [38] (2017)		Ex vivo, bovine livers	4	Sectioning sample and measure ablation zone	10 min, 60 W, 2450 MHz	Longitudinal (L) and transverse (T) RDD *	L: −6.5%, T: −4.0%	
						Advancement	7.4 mm (model) versus 7.5 ± 2.1 mm	
Singh et al. [44] (2019)		Ex vivo, porcine livers	10	Sectioning sample and measure ablation zone	2 min, 40 W, 2450 MHz	Longitudinal (L) and transverse (T) RDD *	L: −13.4% T: 5.4%	Used the experimental results of Wu et al. [52]
Tehrani et al. [46] (2010)		Ex vivo, porcine livers	56	Sectioning sample and measure ablation zone	10 min, 50 and 60 W, 2450 MHz and 80 W, 915 MHz	Longitudinal (L) and transverse (T) RDD *	L: 9%, T: 12%	Used the experimental results of Sun et al. [57]
Wang et al. [52] (2021)		Ex vivo, porcine liver	11	Sectioning sample and measure ablation zone	6 min, 60 W, 2450 MHz	Longitudinal (L) and transverse (T) RDD *	L: 6.8%, T: −4.4%	Used a peristaltic pump to simulate blood circulation and soft plastic tubes for blood vessels
Wang et al. [53] (2023)	54 °C isothermal contour	Ex vivo, porcine liver	5	Sectioning sample and measure ablation zone	8 min, 50 W, 2450 MHz	Longitudinal (L) and transverse (T) RDD *	L: 7.12%, T: 5.56%	Results with 30 mm spacing between antennas
	Arrhenius model					Longitudinal (L) and transverse (T) RDD *	L: −4.98%, T: −13.21%	
Wu et al. [55] (2013)		Ex vivo, porcine livers	10	Sectioning sample and measure ablation zone	2 min, 40 W, 2450 MHz	Longitudinal (L) and transverse (T) RDD *	L: −2.0%, T: 4.2%	

* Relative differences are results of the computational model compared to the experiments. Advancement = the distance from the antenna tip to the boundary of the ablated zone, CT = Computed Tomography, DSC = dice similarity coefficient, MRT = Magnetic Resonance Thermometry, RDD = relative diameter deviation, RVD = relative volume deviation, SAR = specific absorption rate.

**Table 4 cancers-15-05684-t004:** In vivo validation of computational models modeling microwave ablation.

Author (Year)	Model	In Vivo or Ex Vivo Validation	Number of Experiments	Ground Truth	Ablation Settings (Time of Ablation and Power)	Outcome Measure/Metric	Performance	Validation Remarks
Tucci et al. [48] (2022)	Capillaries	In vivo, patients	32	Segmentation on 24 h post-ablation CT	5 and 10 min, 60 W, 2450 MHz	Transverse RDD *	+24% (5 min) +43% (10 min)	Used the experimental results of Amabile et al. [58]
						RVD *	31% (5 min), 93% (10 min)	
	Terminal arteries					Transverse RDD *	−4% (5 min), +8% (10 min)	
						RVD *	−32% (5 min), −8% (10 min)	
	Terminal branches					Transverse RDD * RVD * Transverse RDD * RVD *	−42% (5 min), −43% (10 min)	
							−83% (5 min), −84% (10 min)	
	Tertiary branches						−18% (5 min), −13% (10 min)	
							−88% (5 min), −84% (10 min)	
Zhai et al. [56] (2008)		In vivo, patients	9	Segmentation on 1–2 weeks post-ablation CT	Patient-specific, 2450 MHz	RVD *	±7.0%	Article contains only small details on experiments. Study type unknown

* Relative differences are results of the computational model compared to the experiments. CT = Computed Tomography, RDD = relative diameter deviation, RVD = relative volume deviation.

**Table 5 cancers-15-05684-t005:** Ex vivo validation of computational models modeling radiofrequency ablation.

Author (Year)	Model	In Vivo or Ex Vivo Validation	Number of Experiments	Ground Truth	Ablation Settings (Time of Ablation and Power)	Outcome Measure/Metric	Performance	Validation Remarks
Chang et al. [27] (2004)		Ex vivo, porcine livers	2	Sectioning sample and placed in 2,3,5-triphenyltetrazolium chloride to color cell viability	15 min, 20 V	Longitudinal (L) and transverse (T) RDD *	L: 0.0%, T: 0.0%	
			2		15 min, 25 V		L: −16.7%, T: 20.0%	
			2		15 min, 30 V		L: −4.5%, T: 0.0%	
Chen et al. [28] (2021)	single probe	Ex vivo, porcine livers	5	Sectioning sample and measure ablation zone		Longitudinal (L) and transverse (T) RDD *	L: −0.35%, T: 1.68%	
	Switching probe (10 mm)		5			Longitudinal midline (Lm), longitudinal probeline (Lp), and transverse (T) RDD *	Lm: −1.38%, Lp: −1.82%, T: −0.08%	
	Switching probe (15 mm)		5		12 min		Lm: 0.47%, Lp: 0.05%, T: −0.87%	
	Switching probe (20 mm)		5				Lm: 4.54%, Lp: 0.64%, T: −1.76%	
Duan et al. [32] (2016)		Ex vivo, porcine livers	20	Sectioning sample and measure ablation zone	5 min, temperature-controlled (105 °C)	Longitudinal (L) and transverse (T) RDD and relative area deviation (A) *	L: 11.1%, T:10.9%, A:1%	
Fang et al. [34] (2022)		Ex vivo, bovine livers	3	Sectioning sample and measure ablation zone	12 min, impedance-controlled, 1800 mA	Transverse RDD *	−2.83%	Used the experimental results of Goldberg et al. [59]
Ooi et al. [42] (2019)		Ex vivo, bovine livers	3	Sectioning sample and measure ablation zone	12 min, impedance-controlled, 1800 mA	Transverse RDD *	−20.9%	Used the experimental results of Goldberg et al. [59]
Subramanian et al. [45] (2015)		Ex vivo, bovine livers	15	Segmentation on image of flatbed scanner after sectioning sample	500 KHz, 1–6 min, 31–34 V 60–80 W	Relative area deviation *	−2.63%	Optimized tissue parameters based on experiments
Vaidya et al. [49] (2021)		Ex vivo, phantom	1	Sectioning phantom, using temperature-sensitive ink to measure ablation zone	10 min, temperature-controlled (103 °C), max power of 35 W	Relative area deviation *	17.03%	Used ink which colors irreversibly above threshold T > 70 °C
Wang et al. [51] (2019)		Ex vivo, porcine livers	3	Sectioning sample and measure ablation zone	Temperature-controlled (80 °C), 330 kHz	Longitudinal (L) and transverse (T) RDD *	L: 7.7%, T: 12.8%	Used a peristaltic pump to simulate blood circulation and soft plastic tubes to simulate blood vessels
			3		temperature-controlled (95 °C), 330 kHz		L: 3.9%, T: 21.5%	
			3		Temperature-controlled (90 °C), 330 kHz		L: 0.4%, T: 11.8%	
			3		Temperature-controlled (95 °C), 330 kHz		L: 0.3%, T: 8.1%	
Welp et al. [54] (2006)	Vessel ⌀ = 4 mm, flow 25 mL/min	Ex vivo, porcine livers	10	Sectioning sample and measure ablation zone	12 min, impedance-controlled, 25 W	Transverse RDD *	−5.7%	Used glass tubes to simulate blood vessels
	Vessel ⌀ = 4 mm, flow 50 mL/min						−2.4%	
	Vessel ⌀ = 4 mm, flow 75 mL/min						−8.7%	
	Vessel ⌀ = 6 mm, flow 75 mL/min						1.9%	
	Vessel ⌀ = 6 mm, flow 150 mL/min						1.9%	
	Vessel ⌀ = 6 mm, flow 300 mL/min						1.9%	

* Relative differences are results of the computational model compared to the experiments. CT = Computed Tomography, DSC = dice similarity coefficient, RDD = relative diameter deviation, RVD = relative volume deviation.

**Table 6 cancers-15-05684-t006:** In vivo validation of computational models modeling radiofrequency ablation.

Author (Year)	Model	In Vivo or Ex Vivo Validation	Number of Experiments	Ground Truth	Ablation Settings (Time of Ablation and Power)	Outcome Measure/Metric	Performance	Validation Remarks
Audigier et al. [22] (2013)		In vivo, patients	5 patients, 7 ablations	Segmentation on post-ablation CT scan	Patient-specific	Surface deviation	8.67 mm	Retrospective study
Audigier et al. [23] (2015)		In vivo, patients	10 patients, 14 tumors	Segmentation on post-ablation CT scan	Patient-specific	DSC	0.418	Retrospective study
						Sensitivity	66.94%	
						PPV	38.30%	
Audigier et al. [24] (2017)		In vivo, porcine livers	5 swine, 12 ablations	Segmentation on post-ablation CT scan	6 min, temperature-controlled (105 °C), two iterations for large tumors	Surface deviation	5.3 ± 3.6 mm	Surrogate tumors implanted
						DSC	0.44	
						Sensitivity	47%	
						PPV	53%	
Audigier et al. [25] (2022)	Biophysics-based model	In vivo, patients	11 patients, 12 ablations	Segmentation on post-ablation CT scan	Patient-specific	DSC, surface deviation, and RVD	Best	Retrospective study. Did not express their results numerical, but ranking extracted out of graphs
	Spherical model							
	Eikonal model							
Hoffer et al. [39] (2021)	Computational model	In vivo, porcine livers	2 swine, 6 ablations	Segmentation on post-ablation CT scan		Mean surface deviation	1.1 mm	
						Max surface deviation	5.2 mm	
	Manufacturer’s cart					Mean surface deviation	2.5 mm	
						Max surface deviation	7.8 mm	
Mariappan et al. [40] (2017)	Unknown CT perfusion values	In vivo, patients	6 patients, 10 ablations	Segmentation on 1-month post-ablation CT scan	Patient-specific, temperature-controlled	DSC	0.7286	Retrospective study
						RVD	5.11%	
						Surface deviation	2.55 mm	
	Known CT perfusion values		12 patients, 23 ablations			DSC	0.691	
						RVD	17.93%	
						Surface deviation	2.50 mm	
Moche et al. [41] (2020)		In vivo, patients	46 patients, 51 ablations	Segmentation on 1-month post-ablation CT scan	Patient-specific, temperature-controlled	DSC	0.62 ± 0.14	Prospective study
						Sensitivity	0.70 ± 0.21	
						PPV	0.66 ± 0.25	
						Surface deviation	3.4 ± 1.7 mm	
Payne et al. [43] (2011)		In vivo, porcine livers	2 swine	Segmentation on post-ablation CT scan	Temperature-controlled	RVD	39.6%	
Tucci et al. [47] (2021)	Pennes	In vivo, porcine livers	8 swine	Sectioning sample and measure ablation zone	12 min, 90 V, 500 KHz, impedance-controlled	Transverse RDD *	−32.4%	Compared to experiments of Goldberg et al. [59]
	LTE						−7.57%	
	LTNE						−7.57%	
Voglreiter et al. [50] (2018)		In vivo, patients	21 patients	Segmentation on post-ablation CT scan	Patient-specific	DSC	0.7003 ± 0.0937	Retrospective study
						RVD	13.77 ± 12.96%	
						Sensitivity	69.70 ± 10.94%	
						PPV	71.73 ± 12.00%	
						Surface deviation	2.44 ± 0.84 mm	

* Relative differences are results of the computational model compared to the experiments. CT = Computed Tomography, DSC = dice similarity coefficient, PPV = positive predictive value, RDD = relative diameter deviation, RVD = relative volume deviation.

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
