# Peer review of "Computational Modeling of Thermal Ablation Zones in the Liver: A Systematic Review"

_cancers, 2023, doi:10.3390/cancers15235684_

Round 1

Reviewer 1 Report

Comments and Suggestions for Authors

Authors compare "Thermal" ablations. However, there is no mention of cryoablation or other types of thermal ablation including HIFU and laser ablation. The author should mention that they were only focused on the two more common types of thermal ablation. 

Introduction

line 41, 46 and through out the paper, authors mention "safety margin". This should change to "ablation margin". The minimum of 5 mm around the tumor is not for safety, its to decrease the recurrence rate. 

They can also define: Ablation margin is the area of normal liver around the tumor that is ablated to ensure lower risk of local recurrence.

Author Response

Thank you very much for taking the time to review this manuscript.
Please See the attachment for the response. 

Reviewer 2 Report

Comments and Suggestions for Authors

REVIEW. : computational modeling of thermal ablation zones

The authors have performed a well-written and comprehensive review of the state of the art of computer simulation for liver tumor ablation.   Excellent tabular summarization of the data.

The inability to compare the accuracy of the methods due to variation in reporting and metrics between the various studies warrants further discussion of the metrics utilized.  Are the metrics adequate to assess the accuracy of a simulation?  Even if metrics were common to studies (In 10 or so of the ex vivo MWA models a transverse RDD was available),  these were not analyzed in relation to each other; this warrants additional explanation, perhaps in Discussion regarding optimal study design. 

Regarding the specific metrics, is a relative volume deviation of 0 meaningful if the shapes do not align?.   The metrics should reflect back to the goal of the simulation; to ensure adequate ablation margins are achieved.  While a rough approximation of the true ablation is ensured with overall descriptors (average error or relative volume deviation) (which to some extent are the existing state of the art derived from manufacturer-provided data sheets), the differentiating promise of simulation is its potential to depict the local (patient physiology-specific) variations --  which require attention to the magnitude and prevalence of  maximal surface errors.

Summary : ok

P2: line 53-54:   % of intended margin varied between 2.7% and 51.4%. “These low rates indicate a discrepancy between the predicted and created ablation zone.”  The discrepancy is between the planned or desired ablation zone and that created.  As these are clijnical studies, this may be secondary to the predicted not being that created, but may also reflect an accurate predicted volume being created –but not placed exactly where intended. 

Methods: p3 line 106-107 Considering that the metrics for comparing the models vary, with some similarities, a brief review of what each metric is (and is not) measuring –and to what extent they are comparable--would be useful.   Ie .,in the figure captions, the authors explain that for an experimental diameter of 30 mm a relative error of .1 means the simulated diameter was 33 mm.  This is helpful.  To extend it further, would this, in general, be equivalent to a mean surface deviation of 1.5 mm?  and would a mean surface deviation of 3.4 +/-1.7 mm be expected to indicate that 16% of the surface points had an error of >5.1 mm? 

While averages are useful as overall descriptors of the accuracy of a simulation, when discussing ablation volumes, which may be irregular, the maximum absolute error, and the % of surface subject to that error (or error above some threshhold), are important.

The authors, in focusing on the simulation accuracy, must address its ability to predict a true ablation volume—not only in size and shape, but in a particular position in space.  Most incorporation of simulation entails integration into a more extensive stereotactic 3D multiplanar graphic software.  A key element for accuracy and incorporation in the work-flow, is the method of registration of images and the simulation.  Experimentally, authors may manually co-locate the simulation volume with the ground truth volume (essentially separating the registration issue from the simulation); the method of registration should be explicit. 

--Regarding the metrics, the justification for the use of relative volume deviation (RVD) and relative diameter deviation (RDD) need better explication:  are these both averages that ignore overlap--the position of one volume (simulated) in relation to the other (ground truth)?  (as opposed to DCE.) Similarly, Is RDD calculated similarly for all?; is it taken at target maximum diameter or multiple diameters within the volume?   Ablation volumes are not always regular or symmetric.

Does PPV only indicate the % of voxels in the ground truth volume that were predicted by the simulation?   False positive voxels are also important.

P9 line 145:  “ ..and one used an own heat transfer...” perhaps, ‘..used a proprietary heat ..’

P17 line 294:   this review did not present all characteristics of models.  Time was not included, in particular the time to construct an anatomic model and / or register images, and run the simulation. 

P18. Line 350  ‘... total volume of the simulated and actual ablation are compared without comparison  of the shape and relative position.’   This is a problem as mentioned above—if similar total volumes with dramatically different shapes are highly scored, this is problematic as a metric for an ablation simulation. 

P18 358-360:  “... recommended 5 mm margin, but exact relationship between ablation margins and local recurrence is still unknown [68]”.     (Reference 68 is not listed. --All beyond 57 are missing).

However, though the exact relationship is unclear (ie, a specific  x mm margin has a y % 1year Local tumor progression risk), it is clear from multiple studies in both metastatic and primary liver tumors, that a 5 mm margin for HCC and perhaps a 10 mm margin for CRLM are necessary to reduce local recurrence rates to levels reported for surgical resection.  

And the significance of a 3.4 mm mean surface deviation, as with DICE, is useful for assessing the average performance of a simulation model, but it is the maximum surface deviation that will reveal the accuracy of the simulation to predict whether a circumferential adequate 5mm margin was achieved.  

P18 line 335: “To decide on the optimal balance between complexity, accuracy and computational time, a comparative clinical study should be conducted.” “This validation necessitates a probe position scan, to simulate the ablation at the corresponding probe position as well as a post-ablation CT scan to determine the clinical obtained ablation zone dimensions.”  Not only ablation zone dimensions, but the actual stereotactic location of the ablation zone in relation to the applicator position.  This latter also requires registration of images—which has been problematic in clinical application – and though separate from simulation, should be mentioned as it is integral to its clinical use, and for clinical validation.

P 18 Line 340 :  Agree, the combination of DSC and average and maximum surface deviations—these are probably the optimal metrics.  Though need some quantification of the extent of surface deviations (ie. % of area with deviation > x mm).  The authors have an opportunity to recommend optimal, meaningful, standard metrics for ablation simulation assessment. 

P18 Line 347 :  disagree that “A clinical validation study would also overcome the limitations found in the validation of the included studies.”  Among other things difficult to control, clinical studies would involve more variable registration issues. Is the main limitation of the included studies not more of an issue of using the same metrics?  

Validation of a model, and comparisons between models, can most accurately be performed in an in-vivo study that places an applicator, performs an ablation, generates a simulation, performs a contrast study with imaging at each step under anesthesia, a maximally controlled environment—this should yield comparable data sets to isolate and evaluate the performance of a particular computational model. 

Author Response

(The authors gave the same response as above.)

Reviewer 3 Report

Comments and Suggestions for Authors

The meaning of this paper is not clear. It's certainly not what the title says. Computational modeling of any proces has two important aspects. The first is model of the process which contains physics, chemistry, biology etc. describing it. The second is computational methodology. This text  looks like it was written by the authors who has no interest in either, which is difficult to understand. Also, it is not "systematic revue" - small number of references.

Frankly,  I am very suprised with the sentences

"Studies were identified by searching the electronic databases MEDLINE and Web of 84 Science on April 17, 2023. The search queries were based on synonyms of the keywords 85 “Thermal ablation”, “Liver neoplasm” and “Computational modeling”.

I thought scientific research means years of studying of all relevant sources.

Author Response

Thank you very much for taking the time to review this manuscript.
Please see the attachment for the response. 

Round 2

Reviewer 3 Report

Comments and Suggestions for Authors

The title of the paper shold be changed to 

Computational modeling of thermal ablation zones in the liver: a systematic review of literature